# Raison d'état, Religion, and the Body in *The Rape of Lucrece* †

**Feisal G. Mohamed**

The Graduate Center, City University of New York, 365 Fifth Avenue New York, NY 10016-4309, USA;
fmohamed@gc.cuny.edu

† For their learned and generous remarks on an earlier draft of this essay, I am indebted to my colleagues Mario
DiGangi, Rich McCoy, and Tanya Pollard, as well as to David Urban and this journal's anonymous readers.

**Abstract:** With an emphasis on the religious figuration of its heroine's chaste body, the present essay explores the political dynamics of *The Rape of Lucrece*. The poem draws on Roman religion and Christianity: Lucrece is an emblem of purity, with echoes of the *flaminica* or Vestal virgins, and her spotlessness anticipates Christ's. Seeing these qualities allows us to engage the poem's gender dynamics and its politics, with both of these being centered on issues of property. While *The Rape of Lucrece* has been enlisted as an artifact of late Elizabethan republican culture, its depiction of the expulsion of the Tarquins need not lead us to that conclusion. It is nonetheless a product of the political anxieties of Elizabeth's final years.

**Keywords:** Shakespeare; *The Rape of Lucrece*; republicanism; *raison d'état*; religion; property; body

---

By the measures of the Shakespeare industry, *The Rape of Lucrece* is a work rarely noticed, though the poem's gender dynamics, centered on the rape of its fascinating heroine, have sparked some discussion. Setting the stage for that discussion some forty years ago, Coppélia Kahn perceptively associated Lucrece's suicide with "primitive, nonmoral standards of pollution and uncleanness" characteristic of "the attitudes toward female sexuality underlying Roman marriage".[1] The poem's Roman values have suggested to some that it is a cultural contribution to a late Elizabethan "monarchical republic".[2] In this vein, it has recently been described as favorably imagining a politics where patriarchal power is "not lodged in the body of a single individual but diffused throughout the male citizen body".[3] Andrew Hadfield's excellent book-length study of Shakespeare and republicanism takes the poem's dedication to Southampton, a member of the Essex circle, as a sign of its anti-absolutist spirit, in stride with George Buchanan's account of Tarquin in *De jure regni apud Scotos*.[4] Resistance theory, in Hadfield's telling, is in Shakespeare's moment ineluctably twined with republican thought. Thus the poem's narration of events leading to the establishment of the Roman republic signals an affinity for republican limits on monarchical power.

---

[1] Kahn (1976, p. 49). On gender issues in the poem, see also see Vickers (1985); MacDonald (1994); and Quay (1995). Placing these issues in broader context of the period's various versions of the Lucretia story are MacDonald (1994, esp. 87–89), and Carter (2011, chp. 3), which explores connections between Lucrectia and Philomela. On renditions of Lucretia more generally, see Donaldson (1982); on the rape of Lucretia as figuring humanist enterprise in the period, see Jed (1989).

[2] This influential phrase originates with Collinson (1987).

[3] Kunat (2015, p. 3).

[4] Hadfield (2005, p. 139). Also arguing for the republicanism of the poem is Patterson (1993, pp. 297–311). Colin Burrow persuasively emphasizes the importance of Paulus Marsus' edition of Ovid's *Fasti*, which includes extensive commentary drawing parallels to Livy and other classical sources; see his introduction in Shakespeare (2002, pp. 48–49). Marsus' influence is also noted in Baldwin (1950).

At first glance that reading makes perfect sense, but one must wonder if it really holds up to scrutiny—it is something like determining *Hamlet* to be a play smiling upon Norwegian expansionism. Is a prequel to the advent of the Roman republic necessarily republican in spirit? Certainly the story we get in *The Rape of Lucrece* should make us pause before answering in the affirmative. Though we see the evils of a tyrant's son acting on criminal desires, the expulsion of the Tarquins does not necessarily strike us as leading inexorably to harmonious order. We see, rather, that the republic is built on the ambitions of nobles and relies on the whims of the plebs. And we never see it founded. The poem ends with two perfunctory lines on public assent to the banishment of the Tarquins: "The Romans plausibly did give consent/To Tarquin's everlasting banishment".[5] Colin Burrow remarks that "consent" in the final lines is more complicated than in the poem's argument, coming closer to William Fulbecke's view that the transfer of power from kings to consuls excluded "the people from all right and interest in public affairs".[6] Prepared as Shakespeare was by Livy's and Ovid's versions of the Lucretia story, every early modern reader knows what comes after the expulsion of the Tarquins, but it is striking that the poem does not care to draw our gaze to the breaking dawn of a new order. Livy, by contrast, does not miss the opportunity. In his account, Brutus' vow of revenge is explicitly a vow to end Roman kingship: "I take you, gods, to witness, that I will pursue Lucius Tarquinius Superbus and his wicked wife and all his children, with sword, with fire, aye with whatsoever violence I may; and that I will suffer neither them nor any other to be king in Rome!"[7] Ovid, too, finishes the story on a republican note, with an emphatic statement on the end of kingship: "That day was the last of kingly rule [*dies regnis illa suprema fuit*]".[8] If we want an early modern example, we might look to Thomas Heywood's *The Rape of Lucrece: A True Roman Tragedie*, which ends with Brutus as a consul waging war against the Tarquins.[9] Shakespeare's conclusion is decidedly less republican than that of his main classical sources and that of his contemporary Heywood—for explicit association of the poem's action with constitutional change, we must turn to its argument, which Shakespeare may or may not have written.[10] One furthermore searches in vain for citizens embodying Ciceronian *magnanimitas* in this "republican" poem. The best candidate is Lucrece, who of course does not live to see the end of Roman kingship. Much closer to the surface are the values of *raison d'état*: political prudence and the pursuit of interest. These are certainly the preeminent qualities of Shakespeare's Brutus, who casts aside his disguise and seizes the political opportunity of Lucrece's death: "Brutus who plucked the knife from Lucrece' side . . . Began to clothe his wit in state and pride" (1807–9). Not for the only time in the poem, "state" has multiple registers, here suggesting not only dignity, but a dignity befitting affairs of state. In addition, we soon learn that Brutus had allowed himself to be taken for a fool out of "deep policy" (1815), placing him very explicitly in *raison d'état* thought, in which it is a commonplace that a potential rival keep a low profile during the reign of a tyrant. *Prudentia* comes to have a very prominent place in the poem's narrative development: Brutus' political prudence at poem's end rights the imbalance created by Collatinus' imprudent boasting at its beginning.

Indeed we should expect *The Rape of Lucrece* to be influenced by *raison d'état*, which was very much in vogue at the time of its writing. In studies of English literature, cynical and pragmatic politics

5    Shakespeare (2002, p. 10). Further, parenthetical references to *The Rape of Lucrece* are to this edition, available in Oxford Scholarly Editions online.
6    Burrow introduction to Shakespeare (2002, pp. 46, 73); see also Belsey (2001, p. 334). Burrow and Belsey point to Fulbecke (1601, p. 1): "When vainglorious Tarquine the last of the Romaine kings for the shamefull rape of Lucrece committed by one of his sonnes, was banished from Rome & Consuls succeeded . . . the Romaines changed gold for brasse, and loathing one king suffered manie tyrants". Burrow describes this view as "Tacitean"; though it has debts to the kind of political analysis associated with Renaissance Tacitism, Tacitus himself did look so negatively upon the end of Roman kingship, as is shown below.
7    Livy (1919, pp. 204–05) [1.59.1].
8    Ovid (1931) 2.852.
9    Heywood (1608) sig. K1r. For a recent reading of this play, see Howard (2016).
10   Platt (1975) also notes that the final stanza is "silent about the change in regime," and that the poem's "main and constant theme" is tyranny (64), but nonetheless describes *The Rape of Lucrece* as "republican in sentiment and focus" (p. 76).

are often associated with Machiavelli, but this tends to overlook the broad influence of such writers as Francesco Guicciardini and Giovanni Botero, the latter of whom especially was responding to the broad appetite for *raison d'état* by seeking to clarify and assemble its core principles in a sort of handbook. Guicciardini's history of Italy first appears in English translation in 1579, and Botero's *Della ragion di stato* is first published in Venice in 1589, quickly to become something of a sensation, with further editions in 1590, 1596, and 1598.[11] In addition are the many interpreters of Tacitus, including, most famously, Justus Lipsius, whose *Politicorum* first appeared on the Continent in 1589, was published in London in 1590, and translated into English by William Jones in 1594.[12] This is the same year as the first publication of *The Rape of Lucrece* and both texts were printed by Richard Field, though one wonders how much to make of these coincidences. Maurizio Viroli and Richard Tuck have made clear that *raison d'état* was very much at the center of sixteenth- and seventeenth-century political discourse.[13] And while Hadfield's Essex circle is a hub of republican activity, Tuck just as plausibly frames their politics in terms of revived interest in Tacitus, in which light he views the famous performance of *Richard II* on the eve of Essex's ill-fated rebellion.[14]

　　*The Rape of Lucrece* speaks directly to the concerns animating *raison d'état*, and its particular brand of Tacitism. In the poem's political world *potestas* is divorced from *auctoritas*, leaving us with a skeptical, if not cynical, perspective on the republican politics to emerge from the wreckage of Tarquin's reign.[15] Seeing the poem in this way allows us to bring together conversations on its gender dynamics and its politics: its presentation of a crisis in authority is achieved in no small part through the over-determined figuring of Lucrece's body. The relationship between property and political power is central to that figuring. And religion is fundamental to that figuring, too. We will explore religion as the poem itself demands, in a way that spans Roman religion and Christianity, summoning the resources of both to lend its heroine an aura of purity accentuating the profanity of the political. In this light *The Rape of Lucrece* does not embody a republican spirit so much as it reflects late Elizabethan skepticism on sacred kingship and fears of self-seeking factions awaiting to seize power after the death of the heirless queen.[16]

　　Tarquin the Proud, father of the poem's Tarquin, represents the last in a line of Roman priest-kings, who were not only supreme magistrates but also held the title *rex sacrorum*. Romulus inaugurates this tradition with what Georges Dumézil calls the "trifunctional" nature of ancient kingship, as sovereign, general, and high priest—with formidable learning, Dumézil traces commonality in this respect with Yayati, the "first king" of the Vedic texts, and thence to Iranian and Irish analogues.[17] The religious function of Roman kingship is thought to have become more fully institutionalized with Romulus' successor, Numa, who in his fabled reign of peace and plenty established the temples and rites of Roman religion.[18] Tacitus takes a dim view of Numa's innovations as a form of absolutism, with laws protecting the liberties of the commons becoming possible after the expulsion of Tarquin: "after the absolute sway of Romulus, Numa imposed on his people the bonds of religion and a code dictated by Heaven . . . . Upon the expulsion of Tarquin, the commons, to check senatorial factions, framed a large number of regulations for the protection of their liberties or the establishment of concord".[19]

---

11　Guicciardini (1579); Botero (2017, pp. xxxv–xxxvi).
12　Lipsius (1590, 1594).
13　See Viroli (1992) esp. chp. 6; Tuck (1993) esp. chp. 2. For an illumining overview of *raison d'état* in the period, see Burke (1991).
14　Tuck (1993, pp. 105–60).
15　The reading of *The Rape of Lucrece* here offered grows out of the engagement of *raison d'état* and questions of authority in my *Sovereignty: Seventeenth-Century England and the Making of the Modern Political Imaginary*, forthcoming from OUP.
16　On sacred kingship in the reign of Elizabeth and its relevance to Shakespeare, and *Hamlet* in particular, see McCoy (2002) esp. chp. 1 and 3.
17　Dumézil (1973, pp. 108–10, 119).
18　See Forsythe (2005, pp. 97–101).
19　Tacitus (1931, pp. 562–67) [*Annals*, 3.26–7].

The negotiation between self-interested senators and self-interested commoners is certainly relevant to the picture of Shakespeare's poem that we will paint.

The nature of Roman kingship ought to color how we read the poem's beginning, in which Collatine "unwisely" boasts of his wife's chastity (10). In the poem's terms, the boast is a property claim, and as such carries an inherent challenge to absolute rule: Collatine is not only pointing to his possession of a rare jewel, but also to his confidence in the integrity of his legacy and therefore his possession of Collatium beyond his natural years. Samuel Johnson has such things in mind when declaring the "chastity of women" to be that upon which "all the property in the world depends".[20] Tarquin's "lust-breathèd" flight is described in a way subtly situating Lucrece within these property relationships, first mentioning "Collatium," then "Collatine," and finally "Lucrece the chaste" (3–7). Lucrece's chastity may be the target Tarquin is pursuing, but that target from the very outset is tightly associated with her husband's demesne, an association to be borne in mind when in the next stanza we hear of Collatine's brag: "he, the night before in Tarquin's tent, / Unlocked the treasure of his happy state" (15–16). "State" here takes on multiple meanings, referring to Collatine's general condition, to his marriage, and to his dominion, all of which hinge upon Lucrece's chastity.[21] The last, political sense of the term was one that Shakespeare had used before, in Northumberland's accusation of Richard II: "No more: but that you read / These accusations, and these grieuous Crymes . . . Against the State, and Profit of this Land".[22] Collatium is a commonwealth in miniature, and, given that in this text's frame story all other Roman lords' claims of their wives' chastity have proven to be dubious, it has distinguished itself as a commonwealth of unique integrity. Or, as we will explore more fully, it is a commonwealth that makes present at the site of Lucrece's body the overlapping and tightly associated meanings of integrity: political, moral, and physical.

References to Lucrece as property and polity certainly recur in the text. In a long digression, Tarquin's desire for Lucrece is cast as covetousness (134), and we learn that human striving for "honour, wealth, and ease" can often lead to the kind of "thwarting strife" where one of these is sacrificed for the sake of the other: "As life for honour in fell battle's rage, / Honour for wealth, and oft that wealth doth cost / The death of all" (145–47). As the digression ends and we turn our attention back to Tarquin, covetousness and lust are elided: "Such hazard now must doting Tarquin make, / Pawning his honour to obtain his lust" (155–56). In later casting away his doubts, he likens himself to a merchant worrying about the fate of valuable goods at sea: "Pain pays the income of each precious thing: / Huge rocks, high winds, strong pirates, shelves, and sands / The merchant fears, ere rich at home he lands" (334–36). If Lucrece is figured as valuable property, she is equally figured as a realm vulnerable to conquest. In contemplating his attack on her, Tarquin imagines Collatine rushing home to prevent a "siege that hath engirt his marriage" (221), a language of conquest most pronounced, and given strong erotic charge, as he later gazes on Lucrece's sleeping body:

> Her breasts, like ivory globes circled with blue,
>
> A pair of maiden worlds unconquered:
>
> Save of their lord no bearing yoke they knew,
>
> And him by oath they truly honourèd.
>
> These worlds in Tarquin new ambition bred,
>
> Who like a foul usurper went about
>
> From this fair throne to heave the owner out. (407–13)

A rightful claim, with its fully solemnified founding moment of a marriage vow, is opposed to conquest by force, here styled usurpation motivated by base ambition.

---

20　Boswell (1964, p. 209). This remark, so apposite to *The Rape of Lucrece*, is noted in Kahn (1976, p. 60).
21　See *OED*, 'state,' *sb.* 2, 15.c, 27.
22　Shakespeare, *Richard II*, 4.1.A68-71. All references to Shakespeare's plays are to Shakespeare (2017).

Tarquin's crime is thus cast as both theft and usurpation. But the poem also plays up the fact that Collatine's boasts on Lucrece's chastity do in fact place limits on the power of the house of Tarquin. Early modern questions of absolute and limited monarchy often revolve around the subject's right to hold property. In the absolutist view, the subject cannot make a property claim against the sovereign; such a right obtains only between subjects. "Unto Majestie, or Soveraigntie," declares Jean Bodin, "belongeth an absolute power, not subject to any law," and that absolute power can "dispose of the goods and lives, and of all the state at his pleasure".[23] Though the prince is not bound by human law, he is bound by "the lawes of God and nature".[24] Tarquin, of course, does not feel so bound. Arguments for limited monarchy chip away at this absolute right to dispose of subjects' property. Even Bodin provides a soft form of this limit in stating that a sovereign may not be bound by the law, but is bound by contract, the latter being a matter of private agreement: "the law dependeth of the will and pleasure of him that hath the soveraigntie ... but the contract betwixt the prince and his subjects is mutual, which reciprocally bindeth both parties".[25] Much more radical is the view of the Monarchomachs, which places considerable weight on subjects' property rights in a broader effort to place *lex* above *rex*. François Hotman distinguishes between the king's patrimony and the king's domain: the former "belongs to the king himself" and the king can "alienate it by his own will," whereas "simple ownership of the latter is that of the body of the people as a whole, or of the commonwealth, while the usufruct is the king's [*usufructus autem penes Regem*]".[26] As usufructuary, the king not only cannot alienate the property of his dominion from its rightful owners, the people, but is also obliged to maintain its value. In the *Vindiciae, contra tyrannos*, Brutus goes one step further: since the king does not have absolute title to the fruits of his dominion, so "the title of king does not signify an inheritance, or a property or a usufruct, but a function and a procuratorship". This applies equally to the royal patrimony and to the goods of subjects: "kings are only administrators of the royal patrimony, not proprietors or usufructuaries; ... since this is so, they are clearly still less able to bestow upon themselves the ownership [*proprietas*], the use, or the fruits, either of anyone's private belongings, or of the public belongings of individual municipalities".[27] Should a king seek to deprive a subject of his property without common consent, then he has violated the custodial nature of his office and alienated himself from his claim to lordship. Such arguments on the relationship between property and sovereignty often turn their attention to Rome. In one telling, the *lex regia* records a transfer of sovereignty from the people to the king. In the Monarchomach version, this is not a transfer so much as it is a delegation of authority, so that, as Daniel Lee has shown, the people retain "their collective title of ownership, just as a landlord who leases or mortgages a fief to a tenant still retains the rights of ownership".[28]

The relevance to *The Rape of Lucrece* will be clear. Tarquin seeks to possess and to conquer Lucrece, and, by extension, to disrupt Collatine's title to her and to his dominion. A member of the house of Tarquin flexes in absolutist style two of the three functions of ancient kingship, those of sovereign and of general. But the poem renders such flexion an act of brute force at odds with princely virtue. Absent is the kind of monarchy that would forestall the desire of subjects to limit sovereign power through property claims. Rather, we see everywhere the rapacious attitude toward property characteristic of tyrannous rule, which inevitably provokes a backlash. Tarquin leaves the siege of Ardea, an effort to impose Rome's will on a town hovering between dependence and independence, and, in Livy's

---

[23] Bodin (1606, p. 88). On Lucrece as possession, see also Belsey (2001), esp. 317–19.
[24] Ibid., p. 92.
[25] Ibid., p. 93.
[26] Hotman (1972, p. 255). In Livy's telling, attitudes on property distinguish Tarquin from his predecessor, Servius: where Servius secured his reign by dividing conquered land amongst citizens, thus expanding his base of support, Tarquin used precisely this populism to foment opposition to Servius in the senate; see Livy, *History*, 1.46.
[27] Brutus (1994, pp. 125, 127).
[28] Lee (2016, p. 126).

account, an effort to plunder the wealth of the Rutuli.[29]  And by assaulting and devaluing Collatine's property, Tarquin has proven himself to be an untrustworthy usufructuary.  In their lament after Lucrece's death, Collatine and Lucretius try to outdo each other in their grief over her lost value: "The one doth call her his, the other his, / Yet neither may possess the claim they lay" (1793–94).[30] Under Tarquin the Proud's tyrannous rule they have no means of seeking remedy for the harm they have been dealt, making rebellion the only possible course of action. Shakespeare's presentation of this moment hews closely to Machiavelli's *Discorsi*:

> [Tarquin the Proud] was expelled not because his son Sextus had raped Lucretia but because he had broken the laws of the kingdom and governed it tyrannically, as he had taken away all authority from the Senate and adapted it for himself . . . .  For if Tarquin had lived like the other kings and Sextus his son had made that error, Brutus and Collatinus would have had recourse to Tarquin and not to the Roman people for vengeance against Sextus . . . .  For when men are governed well they do not seek or wish for any other freedom.[31]

In Machiavelli's reading, the tyrant's impulse to run roughshod over law and tradition produces rebellion, as subjects cannot expect appeals to law or tradition to be heard.  And here, as in the passage of Tacitus we have already seen, this lack of good government produces a clamoring after liberties, which can be a public good if also one vexed by competition, instability, and disruption.

The point on good monarchical government receives strong emphasis in the poem.  In her appeals for mercy, Lucrece describes the kind of monarchy at the heart of harmonious order:

> Thou art not what thou seem'st, and if the same,
>
> Thou seem'st not what thou art, a god, a king;
>
> For kings like gods should govern everything.
>
> … …
>
> This deed will make thee only loved for fear,
>
> But happy monarchs still are feared for love.
>
> With foul offenders thou perforce must bear,
>
> When they in thee the like offences prove.
>
> If but for fear of this, thy will remove.
>
> For princes are the glass, the school, the book,
>
> Where subjects' eyes do learn, do read, do look. (600–2, 610–16)

Lucrece is urging Tarquin to embody an idealized monarchical authority.  Strikingly, that monarch is unconstrained by law, but must be cognizant of the example he sets for subjects.  Tarquin's internal government ought to approach god-like impeccability and be the foundation of right rule; it is instead cast in disorder by his unruly "will," a violent appetite which also threatens political disorder. The passage also implies that the king with proper self-government can justly "govern everything," positively imagining enlightened absolutism.  If we style this a republican poem, we overlook such complexities.  That Lucrece is utterly powerless before a monarch who refuses to govern his desires shows how absolute rule shades into tyranny if left unchecked.  That certainly feels like a republican sentiment.  But her expression in this crucial scene of an idealized version of monarchial authority is the most earnest political yearning in the text: "I sue for exiled majesty's repeal" is a desire in the poem that goes unanswered, certainly by Tarquin though also by the political machinations of Brutus (640).

---

[29]　Livy (1919, pp. 196–97) [1.57.1].
[30]　Belsey also notes the emphasis on possession in this moment; see Belsey (2001, pp. 317–18).
[31]　Machiavelli (1996, p. 217) [3.5].

And it feels like more than a rhetorical flourish motivated by her dire situation: we would search the poem in vain for a similarly idealized expression of the virtues of republican government. Republics may secure liberties, especially those centered on property, but authority in the richest sense of the term is attached to a monarch who is a pattern of virtue.

This leads us to consider the third function of kingship, the religious function, by which ancient kings occupied the highest priesthood in the realm. From this quality Tarquin is most thoroughly alienated. Much more than a lapse in princely virtue or a "shame to knighthood" (197), his violent, destructive desires are consistently figured as bestial, even demonic.[32] Lucrece awakes to find Tarquin in her bedchamber and starts as if "she hath beheld some ghastly sprite, / Whose grim aspect sets every joint a-shaking" (451–52). Even as he is contemplating his attack on Lucrece, we are made aware of his distance from religion, which he dismisses outright: "Who fears a sentence or an old man's saw / Shall by a painted cloth be kept in awe" (244–45). Such internal debate between "frozen conscience and hot burning will" is a "graceless" mock of Protestant disputation culture, from which Tarquin further distances himself in anticipating easy, external forgiveness for his crime: "The blackest sin is cleared with absolution" (354). Burrow rightly points out in his edition that "absolution" does not necessarily carry a Roman Catholic charge; the term appears in the general confession of *The Book of Common Prayer*, but there it is God who grants "absolution and remission" of sins. Tarquin seems to be imagining a less exacting, and less penetrating, confessor.

The point here is not that he is a graceless Roman Catholic in a Protestant poem, but rather that the language of grace is deployed to emphasize Tarquin's distance from divine order. And this bears directly on the poem's imagining of political order, because the sacred aura that ought to cling to members of the house of Tarquin is transferred to the chaste body of Lucrece. The contrast could not be more pronounced in the scene of the rape itself:

> Here, with a cockatrice' dead-killing eye,
>
> He rouseth up himself, and makes a pause,
>
> While she the picture of pure piety,
>
> Like a white hind under the gripe's sharp claws,
>
> Pleads in a wilderness where are no laws,
>
> To the rough beast that knows no gentle right,
>
> Nor aught obeys but his foul appetite. (540–46)

In *Amoretti* 49, published the year after *Lucrece*, Spenser more conventionally uses the cockatrice as a figure for his beloved's eyes, with their "powre to kill".[33] Here it takes on a much more sinister charge, reinforced by the sharp claws of the "gripe," most often read as a griffin. Again this is reminiscent of Spenser, who refers to this rapacious mythological beast, and lends it a demonic aspect, in *The Faerie Queene*, with both poets drawing on its traditional association with covetousness.[34] Especially relevant is the vision in Daniel 7, with its four beasts representing iniquitous earthly rule, the first of which is a griffin transformed into a human and whose monstrosity is let loose in the world: "The first was like a lion, and had eagle's wings: I beheld till the wings thereof were plucked, and it was lifted up from the earth, and made stand upon the feet as a man, and a man's heart was given to it".[35] In the vision, the thrones of these beasts are "cast down" to make way for the "Son of man" who emerges

---

[32] Shakespeare will later make the same association in Macbeth's dagger speech: "With Tarquin's rauishing [strides], towards his designe / Moues like a ghost" (2.1.55–56).

[33] Spenser (1989), *Amoretti* 49, 2.

[34] Spenser (2001), *Faerie Queene* 1.5.8, where Sansfoy is likened to a griffin, and 2.11.8, where griffins are among the "misshapen wightes" attacking the House of Alma.

[35] Dan 7.4.

from the "clouds of heaven" and is given "an everlasting dominion, which shall not pass away, and his kingdom that which should not be destroyed".[36]

This resonance with Daniel makes Lucrece's "pure piety" an anticipation of Christ's, an association also made at the moment of her death. In an oddly convoluted image, two streams of blood, red and black, pour from her chest wound at the end of the poem, with the impure blood "that false Tarquin stained" being surrounded by water:

> About the mourning and congealèd face
>
> Of that black blood a wat'ry rigol goes,
>
> Which seems to weep upon the tainted place,
>
> And ever since, as pitying Lucrece' woes,
>
> Corrupted blood some watery token shows,
>
> > And blood untatined still doth red abide,
> > Blushing at that which is so putrefied. (1743–50)

Aristotle had associated black blood with impurity.[37] But the image equally recalls the blood and water draining from the pierced side of Jesus (John 19.34), here adapted to Lucrece in a way equally recalling Saint Augustine's famous reading of her, namely that Christian belief, unlike its Roman counterpart, would offer her a path to redemption even if she had consented to Tarquin's advances.[38] But of course, the presence of the black blood also separates Lucrece from Christ: for all that her chastity is strongly associated with heavenly purity throughout the poem, she is at one remove from Christ's sinlessness. In the poem's thicket of religious signification, Roman and Christian, the corruption of Lucrece is not presented as a merely pagan concern easily transcended by the soul's capacity for Redemption. Her loss of purity cuts deeper.

Lucrece nonetheless remains closer to heavenly purity than are the poem's other characters. That is obvious in Lucrece's appeals to Tarquin, where she most fully embodies the voice of virtue. But it is also true of the fascinating account of her relationship to Collatine. Even as her status as property recurs throughout the poem, Lucrece's inner virtue remains beyond her husband's possession: in Tarquin's tent he "Unlocked the treasure of his happy state, What priceless wealth the heavens had him lent / In the possession of his beauteous mate" (16–18). Dynamics of possession are highly charged in the poem, and we should be attentive to them. Here Collatine possesses his mate, but this treasure's real source of value is possessed by the heavens. Or, put differently, heaven retains ownership of Lucrece's virtue and Collatine is a usufructuary. Just as the poem suggests limits on monarchy in associating Tarquin's encroachments on property as tyrannical and morally corrupt, here limits are placed on Collatine's property rights vis-a-vis his wife's inner beauty. The poem draws a conclusion from Augustine's reading with radical implications in terms of gender and marriage: if Lucrece's soul has an untouchable purity, then that purity must have a heavenly source. Heaven's king cannot be alienated of his property, so Collatine cannot lay claim to Lucrece's virtue. That is not a conclusion in which *The Rape of Lucrece* is especially invested. If marriage does not touch upon internal properties, then the critique intended may take aim at certain interpretations of the marital bond rather than on marriage per se: to be "one flesh" is to be united only in flesh. Here again, as with Tarquin's brief remark on the ease of absolution, heated confessional controversies are suggested rather than fully

---

36　Dan 7.9–12.

37　Aristotle (1965, pp. 218–19) [3.19].

38　See Augustine (1957, pp. 86–87) [1.19]: "she should not have killed herself if it was possible to engage in penance that would gain her credit with her false gods". The claim follows Augustine's critique of the Roman celebration of Lucretia: if she did not consent to Tarquin, then her suicide is the unjust murder of an innocent; and if she did consent, then she should not be celebrated for her purity (pp. 84–87). See also the discussion of the violation of captured virgins on 75–77 [1.16].

engaged: the poem does not want to come down on any particular side but it does seem colored, or at least tinted, by the religious debates with which it is surrounded.

Much more significant is what Lucrece's heavenly virtue implies about Roman religion: if the poem stages the inauguration of the Roman republic, then it also stages a historical moment when the bodies of women come to fullest prominence in Roman religion. Once kings are eliminated, then the tight association of political and religious supremacy in the body of the king falls, too. Coming first to mind will be Vestal virgins, who play a prominent part in the age of the kings and in the republic—according to Plutarch, it is a Vestal virgin who is widely held to have given birth to Romulus.[39] The connection between Lucretia and Vesta was certainly apparent to Middleton, whose 1600 poem *The Ghost of Lucrece* is rife with references to the deity: "Rape . . . hath sepulchred in the shade of dust / *Dianaes* milken robe, and *Vestaes* shield".[40] Even more relevant for our purposes is the *flaminica*, wife of the *flamen dialis*. The republican replacement for the king's status as high priest is the *sacerdos publicus*, an appellation most often applied to the *flamen dialis*. While his office entitled him to a seat in the senate, that right was a later development and not always exercised, and many of the strictures imposed upon him suggest that he must operate at one remove from the dust and heat of daily life, to lend him something of the untouchability of a god. The *flaminica*, selected according to strict rules centered on purity, had a more direct association with sanctifying the business of government and daily life: Macrobius tells us that "it is the custom for the wife of the flamen to sacrifice a ram to Jupiter in the Regia on every market day", which also became days of public business.[41] It is the *flaminica* who enters the space of sovereignty to bless its presiding over economic and legal affairs.

I do not intend to suggest a close, direct parallel between Lucrece and the Vestal virgins or the *flaminica*. But Shakespeare's poem does draw on these politico-religious functions assigned to women in presenting a chaste Lucrece as the character against whom political legitimacy is measured. Awareness of these aspects of Roman civic religion where "pure" women feature prominently lends especial significance to Tarquin's identification of Lucrece as worthy of worship when he is contemplating rape:

> Fair torch, burn out thy light, and lend it not
>
> To darken her whose light excelleth thine;
>
> And die, unhallowed thoughts, before you blot
>
> With your uncleanness that which is divine.
>
> Offer pure incense to so pure a shrine. (190–94)

The more loosely figurative contrast between light and dark shades quickly into a language more pointedly invoking religious ritual—"unhallowed," "uncleanness," "incense," "shrine".[42] In doing violence to Lucrece, Tarquin is doing violence to the entire economy of Roman worship, and the political order to which it is bound.[43] Just as with the issues of property we have explored, we get a sense of Tarquin being alienated from authority before his attack on Lucrece occurs.

It is significant that in calling for revenge, Lucrece is no longer able to invoke her purity in unqualified terms, becoming distanced from the quality on which the Roman religious function of women depends. When she has Collatine and the assembled lords pledge vengeance on her attacker, she is less divine than previously, deploying languages of chivalry and of legal trial: "Knights by their oaths should right poor ladies' harms", she pronounces, and soon after wonders "What is the quality

---

[39]　Plutarch (1914, pp. 96–97) [*Lives*, "Romulus," 3].

[40]　Middleton (1600) sig. B3r. On Vesta, see also Kahn (1976, pp. 50–51).

[41]　Macrobius (2011, pp. 200–1) [1.16.30]. See also Dumézil (1973, pp. 119–24).

[42]　See Burrow n. ad. 192, that "unhallowed" was just coming in the 1590s to develop the sense "wicked" (*OED* 2) as opposed to the stronger sense of "not formally hallowed or consecrated" (*OED* 1).

[43]　In a less religious key, Miola (1983) similarly notes that "Lucrece resides in the middle of the Aristotelian and Ciceronian series of concentric circles that expand outward to include the family, household, city, nation, and world . . . . Tarquin's rape of Lucrece violates all the circles of social order that surround her" (pp. 24–25).

of my offence … May any terms acquit me from this chance?" (1694, 1702, 1706). At this moment the concerns raised by Augustine come to the fore, though that effect is achieved by cleaving to Livy in a way that shows Roman values to be leading Lucrece on the path to self-destruction. In Shakespeare as in Livy, Lucrece asserts her innocence. "Though my gross blood be stained with this abuse," she states in *The Rape of Lucrece*, "Immaculate and spotless is my mind" (1655–56).[44] In both texts the assembled lords agree with this sentiment, and in the poem "all at once began to say / Her body's stain her mind untainted clears" (1709–10). But Lucrece is too much in thrall to a religious and cultural fixation on female purity, as is so visible in Shakespeare's poem when she gazes upon the Troy tapestry and wishes she could tear at Helen's beauty with her nails (1471–72).[45] She imposes the punishment of death on herself so that other women who have been raped do not find mercy: "'No, no,' quoth she, 'No dame hereafter living / By my excuse shall claim excuse's giving'" (1714–15). This again echoes Livy's account, where Lucretia declares, "I do not absolve myself from punishment; nor in time to come shall ever unchaste woman live through the example of Lucretia".[46] The close proximity to Livy throughout this scene makes the more obvious an important departure. When Livy's Lucretia declares herself guiltless, her death is offered as evidence of her innocence: "my body only has been violated; my heart is guiltless *as death shall be my witness* [*mors testis erit*]".[47] Shakespeare's Lucrece states that her soul will "in her poisoned closet yet endure" (1659), playing up the distance between Roman honorable death and Christian heavenly reward. As with the allusion to the griffin of Daniel 7, we are made aware of a Christian dispensation soon to shatter Roman religion like a potter's vessel.

As I have suggested already, it would be terribly reductive to see this elevation of Christianity above Roman religion as the point of the poem. Christianity, rather, enables a spiritual redemption of which Lucrece cannot yet be aware, even as it also disrupts the kind of unified political authority that she, and the poem, seem so strongly to desire. No Christian monarch can be *rex sacrorum* in the pre-republican Roman sense. In addition, until Christ's reign, it is, to return to the language of Daniel 7, our fate to be governed by griffins, flesh-eating bears, winged leopards, and horned beasts, placing enlightened and harmonious politics at one remove from history, in prophetic mode. *The Rape of Lucrece* thus stands as a meditation on the imperfection of political forms, monarchical and republican, which look very much like mechanisms of wielding power. Sometimes these mechanisms are thoroughly corrupt, sometimes modestly so, but, as in the *raison d'état* writers, the pursuit of interest is their lifeblood. Republican settlement, which the poem significantly downplays, does not solve these problems of human government, which are equally visible in Shakespeare's other works of the 1590s, whether on English or Roman history: the *Henry VI* plays, *Titus Andronicus*, *Richard III*, *Richard II*, *King John*, and *Julius Caesar*. In these works, tyranny typically leads to rebellion, but attempts to embody the role of priest-king, visible in Richard II, lead to their own kind of instability. If *The Rape of Lucrece* is not entirely brimming with the spirit of monarchical republicanism, it is nonetheless very much a product of Elizabeth's final years, with their anxieties on transitions of power, their skepticism of a royal authority uniting church and state, and their worries about the kind of interest-driven, factional politics set to emerge from the ashes of a monarchical line.

**Funding:** This research received no external funding.

**Conflicts of Interest:** The author declares no conflict of interest.

---

[44]  Cf. Livy (1919, pp. 202–3) [1.58.7].
[45]  For readings of the Troy tapestry scene, see Benedict J. Whalen's contribution to this special issue, Whalen (2018); see also Maus (1986, pp. 79–82) and MacDonald (1994, pp. 91–96).
[46]  Livy (1919, pp. 202–3) [1.58.10].
[47]  Livy (1919, pp. 202–3) [1.58.7]; emphasis mine.

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
