# Peer review of "Raison d’état, Religion, and the Body in The Rape of Lucrece"

_religions, doi:10.3390/rel10070426_

Round 1

Reviewer 1 Report

This is a strong paper that does a nice job placing some of the religious associations in Lucrece in conversation with the poem's political elements. It is a valuable addition to the special issue. 

A couple of minor suggestions: 

The conclusion that "The Rape of Lucrece thus stands as a meditation on the imperfection of political forms" follows nicely from the arguments of the paper. However, since this is not an entirely new claim, I think some acknowledgement of other, somewhat similar positions (i.e. the work is not simply pro-republican) at some point in the paper would be appropriate. For example, something like Michael Platt's "The Rape of Lucrece and the Republic for Which it Stands" comes to mind. Perhaps some acknowledgement of this sort of push-back against the republican reading of Lucrece in the first or second paragraph of the paper would help. 

On lines 36-39 of the paper, the author makes the excellent point that Shakespeare does not direct our gaze to the new political order, instead ending the poem with those "perfunctory lines". The last lines certainly are perfunctory in the sense the author means, but they are also profoundly appropriate for both the sexual and political themes of the work (and for the general argument of this paper): "plausibly did give consent" raises questions about consent and is less clear about exactly what the will of the people was than the Argument of Lucrece. Colin Burrow has some nice discussion of this element in his introduction to the OUP Complete Sonnets and Poems that might be helpful. I mention this only because I think this actually helps the author's assertions--even right at the very moment of the establishment of the republican government, the poem is somewhat ambiguous about consent, and hence a reader who has just seen Lucrece "yield" because of terror and overwhelming force, must worry about the Roman people as a whole. 

A few style points:

On line 26: "...events leading to [the] establishment..."

On lines 55-56: "here" is repeated awkwardly in the sentence

On lines 100-102: "here" is again repeated awkwardly in the sentence

On line 249: "...to cling to members [of] the house of Tarquin"

Author Response

My sincere thanks to the anonymous reader for this generous and incisive assessment. I am particularly grateful for the suggestion on “consent.” As the reader notes, Burrow’s argument on the conclusion of the poem reinforces my own. I have cited him, and also Belsey’s 2001 article on Lucrece, which makes similar claims about consent in the poem’s final lines.

I am also thankful to have been directed to Michael Platt’s 1975 article on Lucrece, which previously escaped my notice. Though Platt claims that tyranny is the central theme of the poem, he also describes Lucrece as “republican in sentiment and focus” (76; now cited in n10 of the essay). So there is certainly significant daylight between his argument and that here advanced.

Copy-editorial corrections have naturally been made. My apologies that these escaped my notice in the previous draft.

As will be visible in the revised version of the essay, several other sources have now been pulled into the reference list.

Reviewer 2 Report

The essay deal satisfactorily with The Rape of Lucrece, but it requires extensive revision, as indicated in running comments. These pertain principally to style and presentation. The main point of the argument is not made clear at the beginning, where it would be most useful, and it remains unclear throughout. Vague pronoun references are a consistent stylistic problem. I've highlighted as many of these as I noticed, but they're the author's responsibility to identify and correct.

Author Response

My thanks to the anonymous reader. I have responded to running comments in the attached pdf, and changes are tracked in the revised version of the essay.
